# A TRIM insertion in the promoter of *Ms2* causes male sterility in wheat

Chuan Xia[1,*], Lichao Zhang[1,*], Cheng Zou[1,*], Yongqiang Gu[2], Jialei Duan[1], Guangyao Zhao[1], Jiajie Wu[1], Yue Liu[1], Xiaohua Fang[3], Lifeng Gao[1], Yuannian Jiao[4], Jiaqiang Sun[1], Yinghong Pan[1], Xu Liu[1], Jizeng Jia[1] & Xiuying Kong[1]

The male-sterile *ms2* mutant has been known for 40 years and has become extremely important in the commercial production of wheat. However, the gene responsible for this phenotype has remained unknown. Here we report the map-based cloning of the *Ms2* gene. The *Ms2* locus is remarkable in several ways that have implications in basic biology. Beyond having no functional annotation, barely detectable transcription in fertile wild-type wheat plants, and accumulated destructive mutations in *Ms2* orthologs, the *Ms2* allele in the *ms2* mutant has acquired a terminal-repeat retrotransposon in miniature (TRIM) element in its promoter. This TRIM element is responsible for the anther-specific *Ms2* activation that confers male sterility. The identification of *Ms2* not only unravels the genetic basis of a historically important breeding trait, but also shows an example of how a TRIM element insertion near a gene can contribute to genetic novelty and phenotypic plasticity.

[1] Key Laboratory of Crop Gene Resources and Germplasm Enhancement, Ministry of Agriculture, The National Key Facility for Crop Gene Resources and Genetic Improvement, Institute of Crop Sciences, Chinese Academy of Agricultural Sciences, Beijing 100081, China. [2] United States Department of Agriculture-Agricultural Research Service, Western Regional Research Center, Albany, California 94710, USA. [3] State Key Laboratory of Molecular Developmental Biology, Institute of Genetics and Developmental Biology, Chinese Academy of Sciences, Beijing 100101, China. [4] State Key Laboratory of Systematic and Evolutionary Botany, Institute of Botany, Chinese Academy of Sciences, Beijing 100093, China. * These authors contributed equally to this work. Correspondence and requests for materials should be addressed to J.J. (email: jiajizeng@caas.cn) or to X.K. (email: kongxiuying@caas.cn).

Wheat (*Triticum aestivum* L.) is one of the world's most important staple food crops. It provides more than 20% of total food calories for the global population. Current wheat production is not sufficient to satisfy the future demands of the rising global population. Wheat production urgently needs to be increased to ensure global food security[1–3]. The utilization of male sterility is a prerequisite for the economically viable production of hybrid wheat seed and recurrent selection breeding. Therefore, research on male sterility in wheat has long been of great applied significance.

The first dominant male-sterile mutant, Taigu genic male-sterile wheat, which has come to be known as *ms2*, was identified in 1972 (ref. 4). *ms2* mutant plants produce abnormal anthers with no pollen. The male sterility phenotype is conferred by a single dominant gene *Ms2* (refs 5,6), which was found to be located on the short arm of chromosome 4D by cytogenetic analysis[7]. *Ms2* is not linked with any unfavourable agronomic traits and is not influenced by environmental conditions. Use of the *ms2* mutant thus obviated the need for the labour-intensive emasculation step in the production of hybrid wheat seed and, more importantly, accelerated the recurrent selection process in wheat. In the late 1980s, Liu and Yang developed 'Ai-Bai' wheat in which *Ms2* and a dwarfing gene (*Rht-D1c*) co-segregate, resulting in a situation where all of the dwarf plants are sterile and all of the tall plants are fertile[8]. This breeding platform has been widely adopted in wheat breeding programs over the last four decades. By 2009, 42 wheat varieties based on the 'Ai-Bai breeding platform' have been released; these are grown on over 185 million acres and account for 5.6 billion kg of wheat grain production. Although the gene responsible for the phenotypes of *ms2* mutants has been an important target in wheat research for many decades, and despite the achievement of Liu and Deng in locating *Ms2* on the short arm of chromosome 4D (31.16 cM from the centromere) in 1986 (ref. 7), the causal gene responsible for the *ms2* phenotype has remained unknown. Genetic research in wheat is hindered by its tremendous 17-gigabase hexaploid genome, which contains three subgenomes (A, B and D) with high similarity and myriad transposable elements. In the wake of developments in sequencing, a whole genome assembly for bread wheat and the draft sequence of the wheat D-genome progenitor *Aegilops tauschii* and the wheat A-genome progenitor *Triticum urartu* have been released recently[3,9,10]. The draft genome sequences provide the opportunity to improve the resolution of map-based cloning in wheat, especially in the regions with low levels of recombination or polymorphism.

In this study, we clone the *Ms2* gene, the first dominant male-sterile gene, using a positional cloning approach. Using transgenic wheat and ethyl methane sulfonate (EMS)-mediated mutangenesis, we further confirm that the male-sterile phenotype is caused by a terminal-repeat retrotransposons in miniature (TRIM) insertion in the promoter region of *Ms2* gene, which leads to the anther-specific transcription of *Ms2*. We also find defective copies of *Ms2* homologues in the A and B subgenomes and their diploid ancestors. Finally, this TRIM-induced expression of the intact *Ms2* reveals a potential mechanism of gene activation in plants.

## Results

**Map-based cloning of *Ms2*.** The anthers of *ms2* mutant plants degrade completely when the pistil reaches maturity (Fig. 1a,b). Histological analysis showed that the anthers of *ms2* mutant plants developed normally until the early stage of meiosis I prophase (Fig. 1c,d). During meiosis I, around the metaphase stage, the middle layer cells of the *ms2* mutant disappeared early, while the middle layer cells in wild type wheat anther remained intact

(Fig. 1e,f). At the stage when the pollen mother cells of wild-type formed dyads or tetrads, the pollen mother cells and the tapetum of *ms2* mutants degenerated gradually (Fig. 1g,h).

To determine the casual gene responsible for this male-sterile phenotype, we employed a positional cloning approach using a near isogenic line (NIL) segregation population (6,066 individuals) to map *Ms2* to a 165 kb region that contains 14 genes (as predicted by the TriAnnot pipeline; https://urgi.versailles.inra.fr/triannot/?pipeline) (Fig. 1i,j). After sequencing the 14 genes in the candidate regions of sterile and fertile plants in three *ms2* NILs, only one polymorphism, a 1,804 bp DNA insertion in the promoter of the gene no. 14, was consistently present in the male-sterile individuals. We then developed a molecular marker to detect the presence or absence of this insertion in *ms2* progeny from 97 different genetic backgrounds and verified that all fertile plants lacked the insertion while all of the male-sterile plants carried it (Fig. 1k and Supplementary Fig. 1a). RNA-Seq and subsequent polymerase chain reaction with reverse transcription (RT–PCR) analysis showed that the expression of gene no. 14 with the insertion was markedly increased in *ms2* mutant anthers (200-fold) (Fig. 2a and Supplementary Fig. 1b,c; Supplementary Data 1). The DNA sequence at this locus is remarkable in three ways: (1) It contains an 882 bp open reading frame (ORF)-spanning eight exons. The predicted ORF size and exon structures were perfectly aligned (100%) with the full-length cDNA we cloned from the anther of the *ms2* mutant. (2) It has no obvious functional domains according to protein functional domain prediction analysis; it appears to be an orphan gene, with homologues found only in the diploid ancestors of wheat, in tetraploid wheat, and in hexaploid wheat. To further eliminate the possibility that this sequence is a long non-coding gene, we calculated its protein coding potential with the coding potential calculator[11]. Using REF90 in UniProt (ftp://ftp.uniprot.org/pub/databases/uniprot/uniref/uniref90) as a training database, the SVM score from coding potential calculator for gene no.14 is 3.32, which indicates the distance to the hyperplane in the feature space. In a previous description of an lncRNA identification pipeline, sequences with support vector machine (SVM) score <0 were defined as lncRNA[11]. Given that the transcripts with score between −1 and 1 are commonly regarded as 'weak noncoding' or 'weak coding', the possibility of gene no.14 being a protein coding gene appears high. (3) Most striking of all, the 1,804 bp DNA insertion displays the typical structure of the TRIMs[12–14], which belongs to a non-autonomous long terminal-repeat (LTR) retrotransposon family. It contains two identical LTR sequences (582 bp each), two identical 5 bp target site duplications (ACTAG), an internal domain that contains a 19 bp primer–binding site and a 15 bp polypurine tract motif. However, the sequence lacks any coding sequences that would be required for mobility, such as domains characteristic of reverse transcriptase (RT), intergrase or Capsid protein (Fig. 1k).

Since the TRIM element is inserted 309 bp upstream of the transcriptional start site (TSS) of gene no.14, it is conceivable that some combination of the TRIM sequence and the gene sequence could be transcribed as a chimeric gene product. To address this possibility, we generated an artificial wheat genome (Chinese Spring, v0.4.) with this TRIM sequence inserted in front of gene no.14 and mapped all of the RNA-Seq reads from the anther libraries of the *ms2* mutant to this artificial genome. No reads were mapped to this TRIM region or to any position upstream of the TSS, indicating that this TRIM insertion does not affect the gene structure of no.14 and that no chimeric gene product is produced (Supplementary Fig. 2)

**Transgenic validation and EMS mutagenesis analysis of *Ms2*.** To confirm that the male-sterile phenotype was caused by the expression of the candidate *Ms2* gene, the coding sequence of this

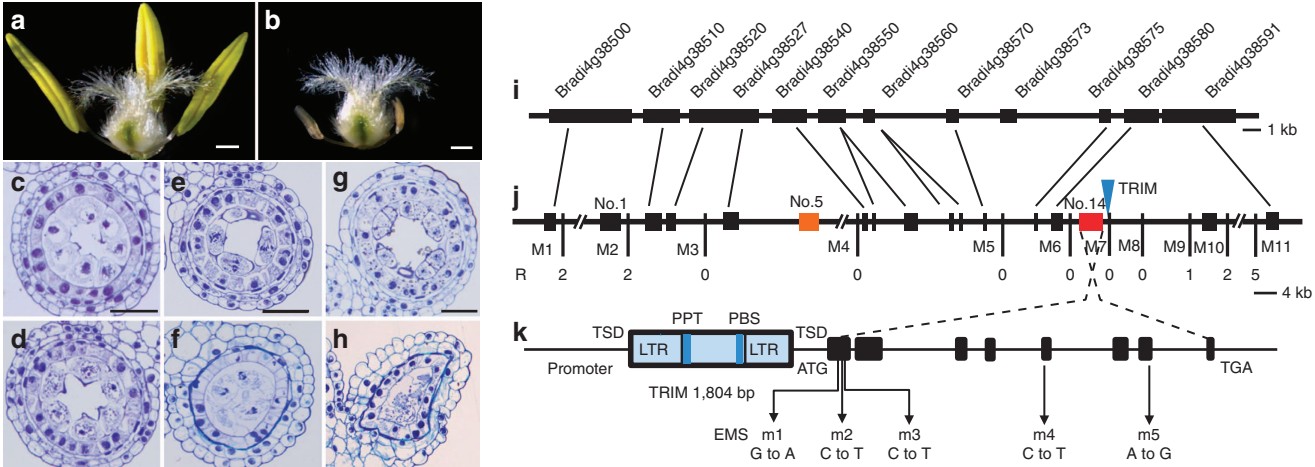

**Figure 1 | Map-based cloning and characterization of *Ms2*.** Phenotypes of floral organs, following the removal of the palea and lemma, of Chinese Spring (**a**) and *ms2* mutant plants (**b**). The anther structure of wild type at the pachytene stage (**c**), metaphase stage (**e**) and telophase stage (**g**); *ms2* anther at the pachytene stage (**d**), metaphase stage (**f**) and telophase stage (**h**). (**i**) The region of *Brachypodium distachyon* chromosome 4 containing 11 genes syntenic with the *Ms2* region of wheat chromosome 4D (**j**), with orthologs indicated by diagonal lines. Vertical lines indicate mapping marker locations, with the number of recombinant chromosomes (R) noted underneath. Gene no.5 (orange) is *PAMs2*, and gene no. 14 (red) is *Ms2*. The blue triangle indicates the TRIM insertion. (**k**) Gene structure of *Ms2*. Black boxes indicate exons and the SNP positions of five EMS mutant lines are indicated and labelled. The blue boxes indicate the TRIM insertion. Bars = 500 μm in **a**,**b**, and 50 μm in **c**–**h**.

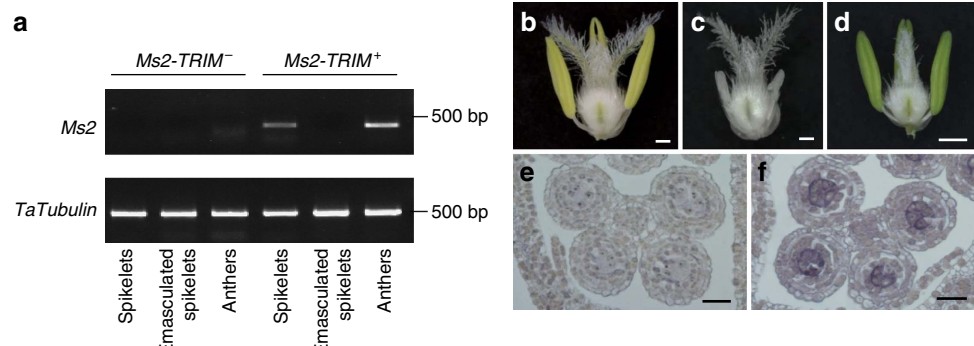

**Figure 2 | Transgenic wheat plants, EMS mutants and gene expression analysis.** (**a**) The RT–PCR result of *Ms2* expression in three reproductive organs (spikelets, emasculated spikelets, anthers) of wild type (*Ms2-TRIM⁻*, fertile) and *ms2* mutant (*Ms2-TRIM⁺*, sterile) plants in the Yanzhan1 genetic background. Phenotypes of floral organs, following the removal of the palea and lemma, of different wheat plants. (**b**) wheat cultivar 'Fielder' transformed with empty vector. (**c**) wheat cultivar 'Fielder' transformed with the **_Ms2-TRIM⁺_**ₚᵣₒ: *Ms2* construct. (**d**) EMS-induced Ai-Bai (*Rht-D1c/Ms2-TRIM⁺*) mutant line m5. *In situ* hybridization analysis (antisense probe) of *Ms2* in cross-sections of prophase stage anthers from wild type (**e**) and *ms2* mutant (**f**) plants. Bars = 500 μm in **b**–**d**, and 50 μm in **e**,**f**.

gene, driven by its native promoter (including the TRIM insertion), was introduced into the wheat cultivar 'Fielder'. Abnormal anthers that had a similar phenotype as *ms2* mutant anthers, were observed in these transgenic wheat plants (Fig. 2b,c), suggesting that this gene does indeed cause the male-sterile phenotype of *ms2* mutant. We also used an EMS-mutagenized population of 'Ai-Bai' wheat to genetically confirm that the phenotype of the *ms2* mutant resulted from the mutation of the candidate *Ms2* gene. A total of 161 of the mutagenized Ai-Bai M₁ plants were fertile and produced normal or larger-than-mutant anthers (Fig. 2d and Supplementary Fig. 3a,b). Sequencing analysis showed that five of these fertile EMS mutants had single nucleotide polymorphisms (SNPs) that resulted in amino acid codon changes in exons of the candidate *Ms2* ORF (Fig. 1k). Moreover, in these five EMS mutant lines, no segregation in fertility was observed in the M₂ generation, indicating that these mutations in *Ms2* are the causal variations responsible for the recovery of fertility.

**The expression pattern of the *Ms2* gene.** RT–PCR and RT-qPCR analysis showed that the expression of the candidate *Ms2* gene was absent or barely detectable in the various tissues of fertile wheat plants, but was specifically present at very high levels in the anthers of male-sterile wheat plants (Fig. 2a, Supplementary Fig. 3c). No mRNA transcripts of the candidate *Ms2* gene were detected in *in situ* hybridization experiments in wild-type plants, but candidate *Ms2* transcripts were present in the anthers of *ms2* mutant plants. The *Ms2* transcripts were concentrated in pollen mother cells in early prophase and were then found in all anthers by late prophase (Fig. 2e,f and Supplementary Fig. 3d,e). To confirm that the anther-specific expression of the candidate *Ms2* can be attributed to the TRIM insertion, we conducted promoter activity assays with green fluorescent protein (GFP) as a reporter. GFP expression driven by the *Ms2-TRIM⁺* promoter was specifically detected in wheat anthers; the *Ms2-TRIM⁻* promoter did not drive any detectable GFP expression (Supplementary Fig. 4a–c). This result supports the idea that this TRIM insertion

results in a novel, anther-specific expression pattern for *Ms2*. Together, our genetic mapping, population genetic analysis, transgenic overexpression, EMS mutant recovery, and organ- and fertility-status-specific mRNA expression analyses strongly support the conclusion that gene no. 14 is the *Ms2* gene, and, further, indicate that the anther-specific expression of this gene is almost certainly responsible for the defective anther development of the *ms2* mutant plants.

**Transcriptome analysis of *ms2* mutant.** To investigate the function of *Ms2* gene, we identified the differentially expressed (DE) genes between the fertile and sterile anthers using RNA-Seq transcriptome analysis. A total of 4,814 genes were found to be differentially expressed between fertile and sterile anthers (a minimum of one fold as the criterion for differential expression, *P* value < 0.01). Among these genes, 665 were expressed to a greater extent in fertile anthers, while 4,149 were expressed to a greater extent in sterile anthers (Supplementary Fig. 5a and Supplementary Data 1). Gene ontology (GO) enrichment analysis indicated that the differentially expressed genes were enriched in 142 biological processes such as 'positive regulation of cell death' (GO: '0010942'), 'sexual reproduction' (GO: '0019953'), 'pollen maturation' (GO: '0010152') (Supplementary Data 2). Studies on anther development in rice and *Arabidopsis* have shown that the orthologs of key regulation genes have similar functions in both species[15,16]. We next checked the homologues of rice anther development-related genes in our wheat transcriptome data and then validated the result using RT–qPCR and RT–PCR. The expression of anther development-related genes, such as *TaMYB80/MYB103* (ref. 17), *TaPTC1* (ref. 18), *Homologue of OsMS2* (ref. 19) and *TaRAFTIN1* (ref. 20), were found to be significantly decreased in the *ms2* mutant (Supplementary Fig. 5b). These results indicated that the *Ms2* might affect anther development through regulating the expression of anther development-related genes.

**Evolutionary trajectory of *Ms2*.** The A, B and D subgenomes in hexaploid wheat are derived from three diploid species. The A genome is originally from the diploid *Triticum urartu*, the B genome is from the diploid *Aegilops speltoides*, and the D genome is from the diploid *Aegilops tauschii*[21,22]. To explore the evolution of *Ms2*, we examined *Ms2* orthologs in all three of the diploid ancestors of wheat and in the hexaploid genome of the wheat cultivar 'Chinese Spring' (Fig. 3a). We found that the *Ms2* orthologs in the genomes of the diploid ancestors have accumulated several destructive mutations. In *T. urartu* (diploid A), the *Ms2* ortholog has a sequence insertion of unknown size in the third exon, as this exon landed in two non-overlapping contigs. In *Ae. speltoides* (diploid B), a stop codon (TAA) caused by a point mutation is present on exon 2. In *Ae. tauschii* (diploid D), we predicted a possible intact ORF and other types of ORFs with different lengths (see the section below). The gene structures of the *Ms2* orthologs in the A and B subgenomes of hexaploid wheat were even more substantially degraded than in the diploid ancestors. In the A subgenome, the *Ms2* homologue has accumulated two deleterious mutations in this coding region; a premature stop codon caused by a frame shift in the first exon and a 9.7 kb insertion in the third exon. Therefore, these deleterious mutations at the *Ms2* locus prevent it from producing a functional protein. In the B subgenome, the *Ms2* homologue is incomplete: only portions of exon5 and exon6 remain, precluding the possibility of encoding a normal protein. A possible intact ORF was predicted for the *Ms2* locus of the diploid D genome. Viewed in combination with the increasing degree of degeneration of *Ms2* from the diploid wheat ancestors to modern hexaploid wheat, these findings suggest that the *Ms2* homologues of the A and B subgenomes have undergone pseudogenization.

To further investigate the evolutionary trajectory of *Ms2*, we sequenced 30 wild diploid D accessions, 28 landraces of hexaploid wheat and 26 modern breeding varieties. While wild *Ae. tauschii* (diploid D) had 14 different haplotypes for the coding region of *Ms2*, the D genomes of the landraces and the modern varieties had only 4 haplotypes (Fig. 3b). These 18 haplotypes can produce 3 types of ORFs, each of different lengths. The longest ORF is 936 bp in length and spans 8 exons (denoted as ORF-I). ORF-II is 852 bp in length and spans 7 exons, and contains a 17-bp deletion that introduces a frame shift that results in a stop codon at the 3′ end that is different from the 3′ stop codon of ORF-I. Given that the *Ms2* orthologs in the A and B diploid ancestor genomes do not have this 17-bp deletion, it is likely that the deletion occurred after the divergence of the A, B and D genomes. This also suggests that ORF-I is more likely to be the ancestor form. For ORF-III, which is present in the *ms2* mutant, it appears that a c.851A > T mutation repairs the ORF-II stop codon, extending the ORF length from 852 to 882 bp spanning 8 exons. Considering that the 17-bp deletion occurs in both haplotype ORF-II and hyplotype ORF-III (which includes the *ms2* haplotype), it is highly likely that the 17-bp deletion happened ahead of the A > T mutation in c.851 of ORF-III. Based on these evolutionary inferences, we propose that ORF-I is the ancestral form, and further propose that ORF-II and ORF-III are derived from ORF-I as a result of multiple evolutionary events (Fig. 3c).

Using the allele spectrum from the D diploid ancestor and from the landraces and modern breeding varieties, we conducted two separate analyses of selection pressure, using both Tajima's D[23] tests and Fay and Wu's D[24] tests. Both analyses indicated that *Ms2* is under neutral selection. However, it must be noted that the landraces had a significantly positive value for Tajima's D, which may be attributable to the shrinking of population size during domestication (Supplementary Table 1). Nevertheless, the neutral selection of *Ms2* is in agreement with the observation that orthologs in the A and B diploid ancestor genomes and the A and B subgenomes of hexaploid wheat have become pseudogenes or are undergoing pseudogenization. Moreover, the *Ms2* gene in the D diploid ancestor genome has become diversified among the 30 wild diploid D accessions.

To explore the emergence of the *Ms2* gene, we searched for its homologues within the plant kingdom. Protein–protein BLAST (blastp) analysis of the NCBI non-redundant protein database and the Ensembl plant protein database found no homologous sequences with an *E*-value < 1. The only paralog we found is the *PAMs2* gene, which shares 31% amino acid similarity with *Ms2* and is located 100 kb away from *Ms2* in the D subgenome (denoted as gene no. 5 in Fig. 1j). Homologues are also present in the A subgenome of hexaploid wheat and in the genomes of *T. urartu* (diploid A) and *Ae. tauschii* (diploid D). These findings indicate that *Ms2* is an orphan gene that is only distantly related to *PAMs2*.

The male sterility phenotype of the *ms2* mutant is triggered by a TRIM insertion. The two LTRs of the *Ms2*-TRIM are identical, suggesting that its insertion occurred very recently. BLAST searching against the Chinese Spring genome assembly (v0.4, https://urgi.versailles.inra.fr/blast) identified multiple hits; the best hit (identity at 96%) is located on the long arm of chromosome 5B (Supplementary Fig. 6). Based on the sequence divergence of the most two similar elements between *Ms2*-TRIM and 5BL-TRIM, we estimate their divergence time to be ~0.96 MYA using a rate of $1.3 \times 10^{-8}$ substitutions/site/year. Therefore, the *Ms2*-TRIM may have originated from an element related to the one in 5BL. BLAST analysis of the TRIM sequences

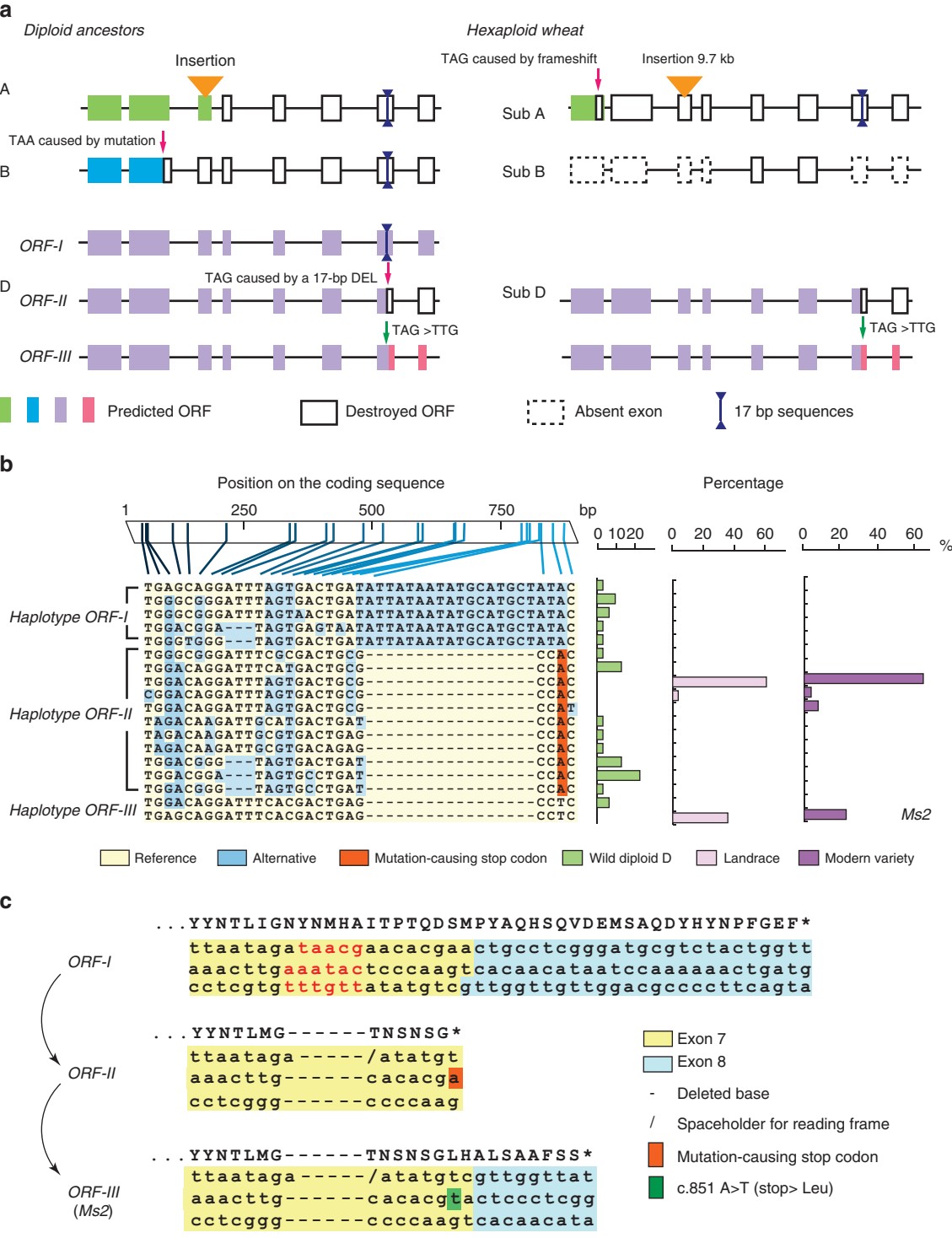

**Figure 3 | Predicted gene structures of Ms2 and haplotype analysis of the coding sequence of Ms2.** (a) Predicted gene structures of the Ms2 loci in the A, B and D diploid ancestor genomes (left panel) and the A, B and D subgenomes of hexaploid wheat (right panel). ORFs are depicted as coloured boxes; the truncated portions of ORFs are depicted as white boxes; red arrows denote the positions of premature stop codons. The orange triangle indicates an insertion in the third exon in the A diploid ancestor genome and in the A subgenome. Exons missing in the B diploid ancestor genome are indicated with dashed boxes. Double arrows in dark blue denote the 17-bp insertion in exon 7. (b) Haplotype analysis of the coding sequence of Ms2. The DNA polymorphisms in the coding sequence are listed at the top. Each column corresponds to a polymorphic site (SNP/Indel), and each row indicates one haplotype of Ms2. The horizontal stacked bar plot indicates, in sequence, the percentage of each haplotype among the wild Ae. tauschii (diploid D) accessions, the hexaploid landraces and the modern wheat varieties. The Chinese Spring (hexaploid 4D) allele of Ms2 was used as the reference (listed in the last row) and is indicated by light yellow cells; blue cells indicate alternate alleles; rust red cells indicate an allele with a premature stop codon. (c) Coding sequences at the end of exon 7 and exon 8 that distinguish ORF-I, ORF-II and ORF-III. The left panel indicates the proposed evolution trajectory of the Ms2 allele in the ms2 mutant.

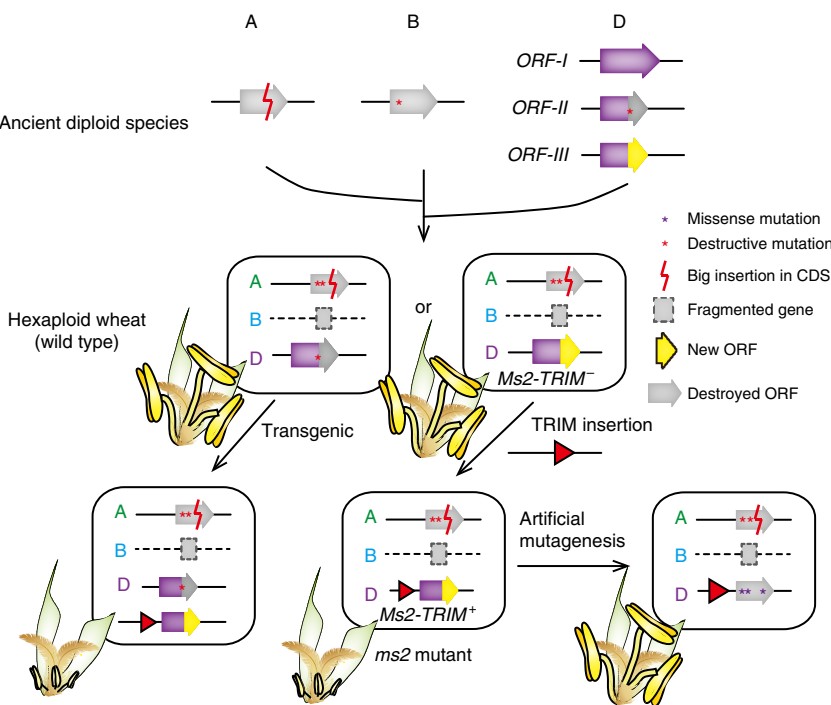

**Figure 4 | The proposed evolutionary trajectory of *Ms2* and a model explaining the fertile and male-sterile phenotypes of different genotypes.** In the A and B diploid ancestor genomes, *Ms2* othorlogs each accumulated one deleterious mutation. Three possible ORFs were predicted for the D diploid ancestor genome. ORF-I was not detected in hexaploidy wheat. In the *ms2* mutant and in the transgenic *Ms2-TRIM*+ wheat (Fielder background), *Ms2* (ORF-III) is activated by the TRIM insertion, leading to male sterility. The EMS-mutagenized mutant lines, which have missense mutations in *Ms2*, produced normal or larger-than-mutant anthers. The purple arrow indicates *Ms2*, the gray arrows indicate truncated ORFs, the yellow arrow indicates the new ORF derived from the frame shift and the subsequent loss of the stop codon. The red triangles indicate the TRIM element in the promoter of *Ms2*.

identified from 48 species[14] (http://bit.ly/1Rtqkie) and in the TREP TRansposable Elements Platform (http://botserv2.uzh.ch/kelldata/trep-db/index.html) did not reveal any similarity with known elements (using *E*-value <1e−5 as the threshold for similarity), suggesting that the *Ms2*-TRIM and 5BL-TRIM belong to a new TRIM family.

## Discussion

Taigu genic male-sterile wheat (*ms2* mutant) is a unique and precious male-sterile genetic resource that has been used widely in wheat breeding programs. We found that it is distinguished from most of the previously identified anther defective mutants in a number of ways. First, it is a dominant and complete male-sterile mutant, which is very rare. Second, the abnormal development of anthers occurs not only in tapetal and pollen mother cells, but in all of the anther cells, including middle layer cells. Third, the degradation of the anther starts at an earlier stage of meiosis than in other mutants, before the microspores have formed.

We here provide several lines of evidence, including genetic mapping, population genetic analysis, transgenic expression, EMS mutagenesis and organ expression analyses, to demonstrate that the *Ms2* gene controls the male-sterile phenotype of the *ms2* mutant (Fig. 4). Moreover, the conspicuous presence of the TRIM insertion in the promoter region of the *Ms2* allele in the *ms2* mutant immediately suggests that this insertion is potentially responsible for the strong expression of the *Ms2* gene in the anthers of sterile *ms2* mutant plants. TRIM elements have been described in plants, ants and tapeworms, and have been proposed to be involved in gene divergence, innovation and regulation[14,25,26]. In plant genomes, TRIM elements are often

close to genes or within genes[12,14], and are thus able to affect the structure and/or expression of such genes. The severe dwarf phenotype caused by the wheat gene *Rht3* (*Rht-B1c*) results from a TRIM insertion in the DELLA domain of the *GAI* gene that alters its coding sequence[27]. Like the *rht3* mutant, the severe phenotype of *ms2* can also be attributed to the TRIM insertion. However, unlike the *rht3* mutant, the TRIM insertion in *ms2* resulted in an anther-specific expression pattern because of its influence on a promoter. Thus, gene function can be significantly affected by TRIM insertions via diverse mechanisms.

Multiple lines of evidence suggest that *Ms2* is a 'dying gene' (that is, a gene probably undergoing pseudogenization). First, when we compared the predicted sequences of *Ms2* in the A and B diploid ancestor genomes with those in the A and B subgenomes of hexaploid wheat, we noted that more destructive mutations have been accumulated in hexaploid wheat. The *Ms2* ortholog in the B subgenome lacks several entire exons. These different states of *Ms2* imply that *Ms2* has undergone pseudogenization during the evolution of the A and B subgenomes (Figs 3a,4). Second, in the D diploid ancestor genome, we characterized three haplotypes based on the size of ORF regions and found that ORF-III (for example, *Ms2* in the *ms2* mutant) is the most recent haplotype. However, at this point, we do not have evidence to show if the gene products of the ORF-I and ORF-II haplotypes have any function. Nevertheless, our results demonstrate that *Ms2* causes male sterility when it is expressed in anther tissues, suggesting that changes in the position of stop codon might not impair its function. Third, population studies of the D diploid ancestor *Ae. taushii* suggest that *Ms2* is under neutral selection, which is common for genes that are undergoing pseudogenization. The observation that the *Ms2* orthologs in the A and B diploid

ancestor genomes and from the A and B subgenomes of hexaploid wheat are fragmented and the fact that none of these genes are expressed further support the idea that the majority of *Ms2* alleles are undergoing pseudogenization. If this is indeed the case, then the TRIM insertion in the promoter of the *Ms2* gene led to the 'resurrection' of a gene that had been undergoing pseudogenization.

In this theoretical *ms2* resurrection scenario, there are two activation steps: the first step is a mutation that continues the translation of a previously truncated ORF (that is, c.851 mutation turning ORF-II to ORF-III); the second step is a TRIM insertion that activates the transcription of this new ORF. The first step is similar to an example of pseudogene resurrection in humans, in which the functional ORF is restored by the loss of three stop codons caused by multiple random mutations[28]. However, in the human example, resurrection of the pseudogene required a long evolutionary time to allow random mutation to rescue the degenerated protein to produce a functional product. Different from the human case, the putative resurrection in *Ms2,* which appears to have occurred in a pseudogene that was not yet fixed in the population. We speculate that this second, 'unfixed' type may be more common in gene evolution than the long-term random rescue of a degenerated protein.

The complex wheat genome, which has undergone multiple rounds of allopolyploidy, has relatively more genes with deleterious mutations than do genomes that have undergone fewer rounds of polyploidization. For example, around 27% of coding loci on the wheat chromosome 3B are likely pseudogenes[29], which is 3 times more than in *Arabidopsis* or rice[30,31]. There must be a large pool of unfixed haplotypes containing deleterious mutations. The identification of the role of *Ms2* in male sterility reveals the genetic basis of a historically important breeding gene. Another important implication from our study is our finding that a TRIM insertion near a gene can contribute to genetic novelty and phenotypic plasticity.

## Methods

**Plant materials.** Our research produced 36 advanced backcross lines that contained the *Ms2* gene in different genetic backgrounds. The *ms2*-CS NIL and *ms2*-Yumai18 NIL were used in phenotype observations. *ms2*-CS, *ms2*-Zhengmai98, *ms2*-Yumai18 and *ms2*-Yanzhan1 NILs were used in the analyses of expression. The segregation population used in the map-based cloning process was derived from crossing synthetic wheat Am3 and *ms2*-Yanzhan1. Ai-Bai wheat was crossed with different germplasm accessions to produce 23 NILs with both the *Ms2* and *Rht-D1c* genes.

The EMS mutants were obtained from EMS mutagenesis of 11,500 hybrid seeds of Ai-Bai wheat. Two EMS concentrations were used; the germination rates of the EMS-mutagenized seeds were 61.5% (0.6% EMS) and 51.5% (0.8% EMS), respectively. A diverse panel lines including of 30 wild relatives (diploid D), 28 landraces and 26 cultivated lines (Supplementary Table 2) were used to detect the haplotypes of the *Ms2* CDS region[32].

The transgenic wheat plants were obtained via an *Agrobacterium tumefaciens*-mediated transformation method using licensed protocols of 'PureWheat' from Japan Tobacco Inc. (JT)[33]. A binary vector (**pLC-Z2**) containing the *Bar* selection gene was used to overexpress *Ms2* in wheat. To obtain transgenic plants, the full-length coding sequence of *Ms2* and the promoter sequence of *Ms2-TRIM*$^+$ were cloned into **pLC-Z2** to create the **$Ms2\text{-}TRIM^+_{Pro}$:Ms2** construct. The plasmids were transformed into the *Agrobacterium tumefaciens* strain EHA105, and introduced into plants of the wheat cultivar 'Fielder'.

**Phenotypic characterization of the *ms2* mutant.** The anthers and pistils were photographed with a stereo microscope (ZEISS-SteREO Discovery.V20). Anther development was observed using semi-thin sections at different developmental stages. The spikelets were collected and fixed in Carnoy's fluid. The samples were dehydrated in an ethanol series and then embedded in Spurr resin (SPI-CHEM). The embedded samples were polymerized at 70 °C and cut into 2 μm sections using a fully motorized rotary microtome (Leica-RM2265). The semi-thin sections were stained with 0.25% toluidine blue and observed under a microscope.

**Molecular cloning of the *Ms2* gene.** The anthers of the mapping population were observed during the flowering stage. A total of 95 polymorphic markers were

identified for fine mapping, using the D genome sequence (http://plants.ensembl.org) and BAC sequences screened from an Ai-Bai wheat library[34]. Gene annotation was conducted using the TriAnnot pipeline (https://urgi.versailles.inra.fr/triannot/?pipeline)[35], using BLAST analysis tools from NCBI (https://www.ncbi.nlm.nih.gov) and using tools available at EnsemblPlants (http://plants.ensembl.org). A large population of *ms2*-Am3 NILs with 6,066 individuals was used for high resolution analysis of the markers. The sequences of markers are listed in Supplementary Table 3.

**In situ hybridization.** Gene-specific primers of *Ms2* (5′-TAATACGACTCACTAT AGGGAGACTGGTGAGGTGTCTCGTGG-3′ and 5′- TAATACGACTCACTAT AGGGAGAGCAGCAGGCAGATAGCAAC-3′) were designed. Probes were transcribed *in vitro* by T7 RNA polymerase, and labelled using DIG RNA labeling Mix (Roche Applied Science). Wheat spikelets of Chinese Spring and the *ms2* mutant at different developmental stages were collected and fixed in formalin-acetic acid-alcohol fixative. The fixed samples were sectioned to 8 μm thickness, and then deparaffinized and rehydrated. Hybridization was performed by using the prepared probes. The fluorescence signal was detected with the antibody conjugated with alkaline phosphatase (Roche Applied Science) and nitroblue tetrazolium chloride/5-bromo-4-chloro-3-indolyl phosphate, toluidine salt (Roche Applied Science). The images were taken via laser confocal microscopy (Leica TCS-SP4)[36].

**Transcriptome analysis.** Total RNA was extracted using TRIzol reagent (Thermo Fisher). The library construction was performed in the following sequences: mRNA enrichment, fragmentation, first strand of cDNA synthesis (random hexamer primers), second strand synthesis (Qiagen), size selection (200 bp) and PCR amplification. The HiSeq 2000 (Illumina) platform was used for the RNA sequencing[37]. Following the filtering of adaptor sequences and low quality regions of the reads, *de novo* assembly was performed according to an optimized pipeline[37], and sequences of less than 200 bp were discarded. Read pairs were aligned against the *de novo* assembled reference by BWA (version 0.7.8), and only uniquely aligned reads were retained for further analysis. Differentially expressed genes were detected using DESeq2 (ref. 38). The assembled sequences were used as queries in BLASTx analysis (version 2.2.30 + ) against the UniProtKB /Swiss-Prot database (release 2014_11). GO terms were retrieved from the best matches. GO term enrichment analyses were implemented using GOseq[39].

For the RT–PCR and RT–qPCR analysis, total RNA was extracted from different wheat tissues using TRIzol reagent. To remove potential genomic DNA contamination, we treated the samples with *DNase* I enzyme (Thermo Fisher). We conducted first-strand cDNA synthesis reaction with 10 μg total RNA using the M-MLV Reverse Transcriptase (Thermo Fisher). For the RT–qPCR experiments, SYBR Premix Ex Taq (Takara) was used. The reactions were performed using the LightCycler 96 system (Roche Applied Science) with the two-step method. The reactions were repeated with three technical replicates of each of three biological replicates. In Supplementary Figs 1b and 3c, the value of $2^{-\Delta CT}$ was used to represent the relative expression. In Supplementary Fig. 5b, the quantitation was performed using the $-\Delta\Delta CT$ method. The primers used in RT–PCR and RT–qPCR are listed in Supplementary Table 3. All of the uncropped gel images are showed in Supplementary Fig. 7.

**Transient expression of GFP in different wheat tissues.** The cauliflower mosaic virus 35S promoter, the *Ms2-TRIM*$^+$ promoter (4,406 bp upstream of ATG) and the *Ms2-TRIM*$^-$ promoter (without TRIM insertion, 2,612 bp upstream of ATG) were fused to *GFP* to generate three transient expression constructs. Different tissues of Chinese Spring wheat were collected and cultured on MS medium, including anthers, pistils, paleas, lemmas, stems and leaves. Three transient expression vectors were bombarded into these wheat tissues using a particle delivery system (PDS-1000/He Biolistic, Bio-Rad). The cells of different tissues were observed via laser confocal microscopy (Leica TCS-SP4) after culturing on MS medium at 25 °C for 16 h.

**Homologous identification of *Ms2* and TRIM.** Protein-protein BLAST searches against the NCBI non-redundant protein database, the Ensembl plant protein database (http://plants.ensembl.org) and WGC assemblies of wheat species (https://urgi.versailles.inra.fr/blast) was conducted to identify the homologues of *Ms2*. There were no hits with an *E*-value < 1 outside the A, B and D genomes of wheat.

The TRIM insertion detection analysis used the dotter[40] and LTR_FINDER (http://tlife.fudan.edu.cn/ltr_finder)[41]. TRIM boundaries and structures were inspected manually. The divergence of two LTRs was calculated using MEGA6 software[42]. The predicted insertion date of a given TRIM element was estimated using the Kimura 2-parameter model[43], with a mutation rate of $1.3 \times 10^{-8}$ substitutions/site/year[44–46].

We searched for the TRIM in *Ms2* throughout the hexaploid wheat genome (http://plants.ensembl.org) using BLAST (blastn). The hits were aligned using MUSCLE[47], and the evolutionary history was inferred by the neighbour-joining method[48]. The total branch length of the optimal tree is 0.036. The evolutionary distances were calculated using the Maximum Composite Likelihood method[49] and

are given in the units of the number of base substitutions per site. Evolutionary analyses were performed in MEGA6 (ref. 42).

**Nucleotide diversity in wild diploid D and wheat accessions.** A total of 84 accessions[32] (30 for wild *Ae. tauschii* (diploid D), 28 landraces of hexaploid wheat, and 26 modern breeding varieties) were used to explore the nucleotide diversity of *Ms2*. The genomic sequences of all samples were successfully amplified by PCR and then sequenced on an ABI 3730 DNA Analyzer (Applied Biosystems). For any polymorphism that occurred only in one accession, we performed the sequencing twice to confirm accuracy. After base calling with ABI Sequencing Analysis software, version 5.3, the data were assembled and manually adjusted using the SeqMan program, version 7.1.0 (Lasergene). The CDS sequences were manually extracted and aligned in MEGA6 (ref. 42).

The diversity levels, measured in samples using Watterson's $\theta^{50}$ and $\pi^{51}$, were calculated using DnaSP[52]. To test if the frequency spectrum of alleles conformed to the expectations of a neutral model, we calculated the value of two statistics: Tajima's D[23], and Fay and Wu's D[24] using DnaSP[52].

**Data availability.** Sequence data supporting this study are available at NCBI GenBank under the accession numbers KX943032, KX943033, KX951468, KY238200-KY238284 and GFFI00000000. The authors also declare that all other data that supports the findings of this study are available within the manuscript and its Supplementary Files or are available from the corresponding authors upon request.

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

## Acknowledgements

This work was supported by grants from The National Key Research and Development Program of China (2016YFD0100302, 2016YFD0101802), the National Transgenic Research Project of China (2009ZX08009-110B), the National Basic Research Program of China (2004CB117202, 2010CB125902) and the CAAS-Innovation Team Project. We thank Y. Liu, L. Zheng, L. Pan, S. Zhang, Y. Jiao, Q. Zuo, J. Xiao, C. Jin, R. Zhou, Z. Guo, X. Guo, C. Xu, X. Zhang, L. Zhang, J. Jiang, Z. Meng, F. Wu, J. Chen, N. Feng, C. Kong and J. Wan for excellent technical support; B. Liu and Z. Wu for providing critical materials; W. Wang, J.H. Snyder, T. Mohr, H. Ma, D. Zhang, Z. A. Wilson, Y. Liu and Y. Zhang for valuable suggestions.

## Author contributions

J.J., X.L. and X.K. initiated the project and designed the study. C.X., L.Z., C.Z., Y.G., J.D., G.Z., J.W., Y.L., X.F., L.G., Y.J., J.S. and Y.P. performed the experiments and data analysis. C.X., C.Z., L.Z., J.J. and X.K. wrote the manuscript, and Y.G., J.S. and Y.J. revised the manuscript.

## Additional information

**Competing interests:** The authors declare no competing financial interests.

