## [Peer Review File · Nature Communications]

Reviewers' comments:

Reviewer #1 (Remarks to the Author):

This research group independently isolated, by map-based cloning, the Ms2 gene. They studied the function of Ms2 by transformation and expression analyses. Further, they investigated the evolution of Ms2 in cultivated and wild wheat species. Ms2 is a new gene activated in anthers by a transposon insertion in the promoter region of a silent gene, and its dominant function is to cause male sterility. The results show how a silent pseudogene is resurrected to an active functional gene. Since this work has complementary results with that by Fu et al. describing this same gene, I agree with co-publication of the two works.

Major comments:

The description of the evolutionary trajectory of Ms2 in the *Triticum/Aegilops* genus is not clear. With my understanding, the evolutionary trajectory of Ms2 should be as follows: (1) Duplication/rearrangement events from the active ancient gene PAMs2 in a common ancestor of the *Triticum/Aegilops* genus produced the linked primary pseudogene (I); (2) During the evolution of the ancient diploid *Triticum/Aegilops* species with A, B, and D genomes, the diverged sequences Ψ Ms-A2 (pseudogene II), Ψ Ms-B2 (pseudogene III), and Ms-D2 (pseudogene I) were produced, respectively. (3) In the hexaploid wheat, these three types of pseudogene sequences are present together in the genomes. (4) A nucleotide mutation in Ms-D2 of hexaploid wheat, which produced a premature stop codon, generated another type of pseudogene IV. (5) A TRIM insertion in Ms-D2 of hexaploid wheat caused the active gene Ms2 (Ms2-TRIM+).

Other minor comments"

1) Figure 1 legends: (d) 5), The recombinant number should indicate "the number of recombinant chromosome", not "the number of recombinant plants" ("recombinant plant numbers (RN)"), because a homozygous recombinant plant has two recombinant chromosomes, while a heterozygous recombinant plant has one recombinant chromosome. Thus, use of "recombinant plants" is confusing.

2) Figure 1I and Supplementary Figure 1I. The relative expression values should be shown as the direct ratios of the target gene to the referent control gene (the control gene also should be indicated). Because the wild-type (Ms2-TRIM-) gene is silent (or very low expression), converting the very low value (probable noise signal) of the Ms2-TRIM- gene as 1 and the values of the functional Ms2-TRIM+ to the ratios of Ms2-TRIM+ / Ms2-TRIM- is not reasonable; which produce very large differences among independent experiments, because the fold changes of the low values for Ms2-TRIM- among independent experiments can be very large.

3) More detailed description in legend of Supplementary Fig. 2a should be given.

Reviewer #2 (Remarks to the Author):

This manuscript describes the molecular characterisation of the wheat Male Sterile 2 gene (Ms2) and convincingly demonstrates that the causal agent is increased expression in anthers caused by an insertion of a TRIM element into the promoter of the D genome Ms2 homeoallele. This is clearly shown and articulated, and the conclusions are supported by the data.

The authors then discuss the evolutionary history of the Ms2 genes on different genomes and in the diploid wheat ancestors. This part of the manuscript is very interesting and provides evidence to suggest that all/some of the Ms2 genes (on the A,B,D genomes) have become pseudogenes, and that resurrection of the D version Ms2 homeoallele has given rise to the male-sterility inducing form by novel expression due to the TRIM insertion.

Point-by-point response to the referees' comments:

Reviewer #1

Major comments:

The description of the evolutionary trajectory of *Ms2* in the Triticum/Aegilops genus is not clear. With my understanding, the evolutionary trajectory of *Ms2* should be as follows: (1) Duplication/rearrangement events from the active ancient gene PAMs2 in a common ancestor of the Triticum/Aegilops genus produced the linked primary pseudogene (I) ; (2) During the evolution of the ancient diploid Triticum/Aegilops species with A, B, and D genomes, the diverged sequences *ΨMs-A2* (pseudogene II), *ΨMs-B2* (pseudogene III), and *Ms-D2* (pseudogene I) were produced, respectively. (3) In the hexaploid wheat, these three types of pseudogene sequences are present together in the genomes. (4) A nucleotide mutation in *Ms-D2* of hexaploid wheat, which produced a premature stop codon, generated another type of pseudogene IV. (5) A TRIM insertion in *Ms-D2* of hexaploid wheat caused the active gene *Ms2* (*Ms2-TRIM+*).

Response: We really appreciate the reviewer's constructive and extremely clear/concise suggestion. We have summarized our result into a new figure according to the reviewer's suggestion. This provides a better illustration of the evolutionary trajectory of *Ms2* and how the phenotype determined by the genetic factors.

Minor comments

1) "Figure 1 legends: The recombinant number should indicate "the number of recombinant chromosome", not "the number of recombinant plants" ("recombinant plant numbers (RN)"), because a homozygous recombinant plant has two recombinant chromosomes, while a heterozygous recombinant plant has one recombinant chromosome. Thus, use of "recombinant plants" is confusing. "

Response: Thank you for this comment. We have already changed the "recombinant plant numbers (RN)" into "the number of recombinant chromosomes (R)".

2) "Figure 1I and Supplementary Figure1I. The relative expression values should be shown as the direct ratios of the target gene to the referent control gene (the control gene also should be indicated). Because the wild-type (*Ms2-TRIM-*) gene is silent (or very low expression), converting the very low value (probable noise signal) of the *Ms2-TRIM-* gene as 1 and the values of the functional *Ms2-TRIM+* to the ratios of *Ms2-TRIM+* / *Ms2-TRIM-* is not reasonable; which produced very large differences among independent experiments, because the fold changes of the low values for *Ms2-TRIM-* among independent experiments can be very large."

Response: We totally agree with the reviewer's comments. The classic way of using the comparative CT method ($-\Delta\Delta Ct$) to indicate the relative gene expression level is not appropriate in our case. The $\Delta\Delta Ct$ is determined by the formula: $\Delta\Delta Ct = (Ct_{\text{target gene}} - Ct_{\text{reference gene}})_{\text{experiment}} - (Ct_{\text{target gene}} - Ct_{\text{reference gene}})_{\text{control}}$. A normalized value of the expression level relative to the calibrator (R) is $R = 2^{-\Delta\Delta Ct}$. In this study, the R is extremely large because the gene expression in *Ms2-TRIM* is very

low. Therefore, instead of using the $-\Delta\Delta C_t$ method, we use the value of $2^{-\Delta C_t}$ to represent the relative expression value. The relative expression figures have been replaced with new one in Supplementary figure 1b and 2c. In order to make it easier to comprehend, we used the raw RT-PCR gel electrophoresis to show the expression of *Ms2* in this version. The RT-PCR gel electrophoresis clearly indicated that compared to the control gene (*TaTubulin*), *Ms2* was expressed at a much higher level. Here we also attached the raw C_t and ΔC_t of different tissue expression analysis in the following table for your reference.

sample	Ms2 Average C_T	GAPDH Average C_T	ΔC_T (Ms2-GAPDH)	$\Delta\Delta C_T$ ($\Delta C_T - \Delta C_T$ of anther)	relative expression $2^{-\Delta\Delta C_t}$
spikelets of Ms2-TRIM ⁻	31.35±0.43	17.85±1.23	13.5±0.8	-2.91±0.02	7.49±0.16
emasculated spikelets of Ms2-TRIM ⁻	30.07±0.05	18.59±0.36	11.48±0.32	-4.93±0.46	32±13.95
anthers of Ms2-TRIM ⁻	33.28±0.32	16.87±1.1	16.4±0.78	0	1
spikelets of Ms2-TRIM ⁺	28.7±1.13	21.05±2.22	7.69±1.09	-8.71±0.31	429.86±129.58
emasculated spikelets of Ms2-TRIM ⁺	32.43±0.57	18.73±1.2	13.7±0.63	-2.7±0.15	6.55±0.96
anthers of Ms2-TRIM ⁺	22.72±0.05	18.33±0.66	4.39±0.71	-12.02±0.07	4143.46±277.42

3) “More detailed description in legend of Supplementary Fig. 2a should be given.”

Response: We are sorry that we did not provide enough information in our figure legend. A detailed figure legend about how this analysis was conducted and descriptions of every symbol have been added in this version.

Reviewers' comments:

Reviewer #3 (Remarks to the Author):

Review on Xia et al. "A TRIM insertion led to a gene resurrection event that causes male sterility in wheat"

The article from Xia and colleagues describes the map-based cloning of Ms2 gene involved in male sterility in wheat. The study is sound and the results are novel. It provides strong evidence for the role of the recent insertion of a non-autonomous retrotransposon (TRIM) in the overexpression of the downstream gene and in the male sterile phenotype. However, some important details remain unclear to me, so please see below my concerns about the evolutionary scenario and about data interpretations.

- Line 97: gene prediction:

There are not enough details to understand how this gene#14 was predicted, based on which kind of evidence. From what I have read, this locus shares similarity with transcript (RNASeq) in ms2 mutant but the transcribed region does not share any significant similarity with any known proteins. It is mentioned that "It contains an ORF" but ORFs are present everywhere so it is not a strong argument to validate the presence of a protein-coding gene. It depends on the size of the ORF at least. So how did you decide this locus is protein-coding or noncoding (lncRNA)? This has strong implications in the definition of pseudogene actually! so it is definitely critical for the manuscript which concludes to the "reactivation of a pseudogene".

- line 164: "we found an allele" at codon 851 "that introduces a premature stop codon". How many codons are present in the predicted CDS? From Figure 3b, I guess around 855-860. So, should you really consider the introduction of a "premature" stop codon at position 851 as inactivating mutation? The probability that the introduction of a stop codon at the 3' extremity of the CDS (probably <10 codons away from the original stop) may inactivate the function of the protein is low to my opinion. And in addition, admit the gene is protein-coding, since the gene is an orphan, one may consider the gene model is not of high confidence. For instance, what is called the "premature stop codon" (c851) could be the original stop codon actually.

- Lines 173-178 and Fig 3: "allele frequency of the intact versus pseudogene alleles"
Since I doubt there is a functional versus a pseudogene allele here, I suggest modifying this part.

- Conservation between PAMS2 and Ms2:

What is the level of conservation between the two predicted CDS/proteins? Is it in the range expected for a pair originating from a duplication event in the progenitor of Aegilops/Triticum? From what I see in Supp Fig4, the Ks is between 0.65-0.75 which looks to me quite high for such a recent event. I would expect a very low Ks if it's a recent duplication).

The hypothesis of a recent duplication event, and the "relaxed" selection compared to

"genes that diverged at about the same time" is not convincing to me, so I suggest either correcting or giving more arguments to support it.

- The resurrection of a pseudogene.

Lines 204-211: the 96% "similarity" should be corrected to 96% "identity" (similarity is for protein alignments). "Translocation" should be corrected to "transposition". 96% identity is low actually to believe the copy on chromosome 5BL is the copy that has transposed. At such short evolutionary distance, one would expect 99-100% identity, or one should consider an accelerated evolution. I suggest to discuss this point.

- Lines 212-224: the evolutionary model

I suggest modifying this part according to previous remarks, or giving more convincing evidence.

What is convincing in this study is that the ms2 phenotype is related to the overexpression in the anther of a gene that has originated by the recent insertion of a TRIM element in the promoter region. This is a very interesting result and it is the main message of the manuscript.

The interpretation of the data as "pseudogene resurrection" appears somehow too speculative to me. This pseudogene is defined as pseudogene only because no expression was detected, which is questionable to me. It would have been different if one can identify inactivating mutations (like on subgenomes A and B). Expression of Ms2 could simply be specific to a certain tissue or developmental stage for which there is no transcriptomic sample available for the moment. So, my feeling is that the resurrection of a pseudogene is still a hypothesis here, and I suggest describing it as a hypothesis rather than as a demonstrated result (especially as written in the title). The gene might be simply silent in the anther and the consequence of the TRIM insertion is the activation of transcription in a tissue where the gene is usually silent.

- TRIM mediated-expression: I think additional details could have been provided regarding the expression of this silent gene from a TRIM. Since RNASeq data are available, could you please comment on the transcript that is generated here? Is the transcript expressed from the promoter in the LTR of the element (=chimeric transcript TE-gene)? Or is the TSS (transcription start site) outside the TRIM?

- Methods: Identification of the TRIM:

LTR_Finder is a good program to predict the presence and structure of LTR-RTs. But the predictions are often incorrect in the wheat genome especially because of nested insertions that can split LTRs into several pieces. It requires manual curation to validate the borders of the element (using dotter for instance).

- Figure 1: TDR should be corrected to LTR.

Reviewers' comments:

Reviewer #3 (Remarks to the Author):

Review on Xia et al. "A TRIM insertion led to a gene resurrection event that causes male sterility in wheat"

The article from Xia and colleagues describes the map-based cloning of *Ms2* gene involved in male sterility in wheat. The study is sound and the results are novel. It provides strong evidence for the role of the recent insertion of a non-autonomous retrotransposon (TRIM) in the overexpression of the downstream gene and in the male sterile phenotype. However, some important details remain unclear to me, so please see below my concerns about the evolutionary scenario and about data interpretations.

Comment 1

- Line 97: gene prediction:

There are not enough details to understand how this gene#14 was predicted, based on which kind of evidence. From what I have read, this locus shares similarity with transcript (RNASeq) in *ms2* mutant but the transcribed region does not share any significant similarity with any known proteins. It is mentioned that "It contains an ORF" but ORFs are present everywhere so it is not a strong argument to validate the presence of a protein-coding gene. It depends on the size of the ORF at least. So how did you decide this locus is protein-coding or noncoding (lncRNA)? This has strong implications in the definition of pseudogene actually! so it is definitely critical for the manuscript which concludes to the "reactivation of a pseudogene".

Response:

We are sorry that we did not provide enough details about the *Ms2* annotation. There are multiple lines of evidence indicating that the *Ms2* is a protein coding gene rather than an lncRNA. First, we used the Coding Potential Calculator (CPC) software tool to evaluate the protein coding potential of *Ms2* gene. Using REF90 in uniprot (<ftp://ftp.uniprot.org/pub/databases/uniprot/uniref/uniref90>) as a training database, the SVM score from CPC for *Ms2* is 3.32, which reflects the 'distance' to the SVM classification hyper-plane in the features space. In a previous description of an lncRNA identification pipeline, sequences with SVM score less than zero were defined as lncRNA. Given that the transcripts with score between -1 and 1 are defined as 'weak noncoding' or 'weak coding' according to the parameters used in the CPC program, the possibility of *Ms2* being a protein coding gene appears high. Second, *Ms2* has been annotated in public databases as a high confidence protein coding gene in the Ensembl Plants website (<http://plants.ensembl.org>), the gene no.14 has been annotated as TRIAE_CS42_4DS_TGACv1_361189_AA1163110. The 1304bp full-length cDNA sequence, the 882bp CDS, and 293aa protein sequence of this gene have been given in detail. In addition, we recently constructed a prokaryotic expression vector using the *Ms2* CDS, and the fused protein with a GST tag could be expressed and purified in *E. coli*.

Considering these multiple lines of evidence, the original *Ms2* seems highly likely to be a protein

coding gene rather than an lncRNA.

Comment 2

- line 164: "we found an allele" at codon 851 "that introduces a premature stop codon".

How many codons are present in the predicted CDS? From Figure 3b, I guess around 855-860. So, should you really consider the introduction of a "premature" stop codon at position 851 as inactivating mutation? The probability that the introduction of a stop codon at the 3' extremity of the CDS (probably <10 codons away from the original stop) may inactivate the function of the protein is low to my opinion. And in addition, admit the gene is protein-coding, since the gene is an orphan, one may consider the gene model is not of high confidence. For instance, what is called the "premature stop codon" (c851) could be the original stop codon actually.

Response:

We agree with the reviewer that we did not describe our result precisely. The predicted CDS of *Ms2* is 882 bp (293 aa). We have amplified and sequenced the 882 bp CDS from the *ms2* mutant anther, which was confirmed by our *de novo* assembly of RNA-seq data (The Transcriptome Shotgun Assembly project has been deposited at DDBJ/EMBL/GenBank under the accession GFFI00000000). In the revised manuscript, we have provided more information to support the notion that c.851 is unlikely to be the original stop codon.

The stop codon at c.851 is caused by a 17-bp deletion that introduces a frame shift. Given that the *Ms2* orthologs in the A and B diploid ancestor genomes do not have this 17-bp deletion, it is likely the sequence without this 17-bp deletion is the ancestor form (we have denoted it as ORF-I in the manuscript). As we illustrated in the updated Figure 3, there are three possible ORFs among the *Ms2* alleles in the diploid ancestor D genome. The longest ORF (denoted as ORF-I) is 936 bp in length and spans 8 exons. ORF-II is 852 bp in length and spans 7 exons, and contains a 17-bp deletion that introduces a frame shift that results in a stop codon at the 3' end that is different from the 3' stop codon of ORF-I.

For ORF-III, which is present in the *ms2* mutant, it appears that a c.851A>T mutation repairs the ORF-II stop codon, extending the ORF length from 852 to 882 bp spanning 8 exons. Considering that the 17-bp deletion occurs in both haplotype ORF-II and haplotype ORF-III (including the *ms2* haplotype), it is highly likely that the 17-bp deletion happened before the event of the A>T mutation in c.851 of ORF-III. Based on these evolutionary inferences, we propose that ORF-I is the ancestral form, and further propose that ORF-II and ORF-III derived from ORF-I through multiple evolutionary events (Fig. 3c).

Although it is reasonable to deduce that the c.851 stop codon (31 aa away from the original stop) is a 'premature' stop codon, we agree with the reviewer that this stop codon may not inactivate the

protein function. Therefore, we have remove the description of ‘premature stop codon’.

Comment 3

- Lines 173-178 and Fig 3: "allele frequency of the intact versus pseudogene alleles"

Since I doubt there is a functional versus a pseudogene allele here, I suggest modifying this part.

Response:

Yes, we agree that the pseudogene resurrection hypothesis is only one of many possible scenarios that might have happened. And the functional study of the proposed intact copy is hard to perform owing to the lack of expression and the lack of any predicted functional domains; therefore, we modified our manuscript intensively according to your suggestion. All of the statements regarding pseudogenes have been modified in the revised text (we elaborate on this issue in further responses, below).

Comment 4

- Conservation between PAMS2 and Ms2:

What is the level of conservation between the two predicted CDS/proteins? Is it in the range expected for a pair originating from a duplication event in the progenitor of Aegilops/Triticum? From what I see in Supp Fig4, the Ks is between 0.65-0.75 which looks to me quite high for such a recent event. I would expect a very low Ks if it's a recent duplication).

The hypothesis of a recent duplication event, and the "relaxed" selection compared to "genes that diverged at about the same time" is not convincing to me, so I suggest either correcting or giving more arguments to support it.

Response:

The similarity between these two predicted proteins is about 31%. We agree with the reviewer that we do not have strong evidence to infer the direct correlation between *PAMs2* and *Ms2*, as the sequence similarity is low and lacking in other supporting evidence. In the revised text, we simply indicate that *Ms2* is distantly related to *PAMs2*.

Comment 5

- The resurrection of a pseudogene.

Lines 204-211: the 96% "similarity" should be corrected to 96% "identity" (similarity is for protein alignments). "Translocation" should be corrected to "transposition". 96% identity is low actually to believe the copy on chromosome 5BL is the copy that has transposed. At such short evolutionary distance, one would expect 99-100% identity, or one should consider an accelerated evolution. I suggest to discuss this point.

Response:

Thanks for the suggestions; we have corrected the ‘similarity’ to ‘identity’ and deleted ‘translocation’ in the revised version.

Based on our BLAST search, *Ms2*-TRIM is most closely related to the copy on chromosome 5BL.

As pointed out by the reviewer, 96% identity between *Ms2*-TRIM and 5BL-TRIM is too low to believe that the *Ms2*-TRIM is directly derived from the copy on chromosome 5BL. We estimate their divergence time to be ~0.96 MYA using a rate of 1.3×10^{-8} substitutions/site/year. Therefore, the *Ms2*-TRIM may have originated from an element related to the one in 5BL. One can argue that TRIM element which transposed to *Ms2* might have been lost or diverged in the genome, or may not be covered by the present wheat reference genome.

In addition, we performed BLAST analysis for the *Ms2*-TRIM and 5BL-TRIM against the TRIM sequences identified from 48 species (<http://bit.ly/1Rtqkie>) and in the TREP (TRansposable Elements Platform) (<http://botserv2.uzh.ch/kelldata/trep-db/index.html>), we did not find any hits (using E-value <1e-5 as the threshold), suggesting that the *Ms2*-TRIM and the 5BL-TRIM belong to a new TRIM family.

Therefore, we changed the description of ‘5BL-TRIM directly transposed to *Ms2* promoter’ into ‘the *Ms2*-TRIM and 5BL-TRIM belong to a new TRIM family’.

Comment 6

- Lines 212-224: the evolutionary model

I suggest modifying this part according to previous remarks, or giving more convincing evidence.

Response:

Thanks for your suggestion. We have cut down the content about the evolutionary model significantly and have moved it to the discussion. We have provided more details about the different haplotypes of *Ms2* in the results and considered their possible evolutionary trajectory. In the discussion, we have stated the hypothesis that we proposed and modified the evolutionary model accordingly.

Comment 7

What is convincing in this study is that the *ms2* phenotype is related to the overexpression in the anther of a gene that has originated by the recent insertion of a TRIM element in the promoter region. This is a very interesting result and it is the main message of the manuscript.

The interpretation of the data as "pseudogene resurrection" appears somehow too speculative to me. This pseudogene is defined as pseudogene only because no expression was detected, which is questionable to me. It would have been different if one can identify inactivating mutations (like on subgenomes A and B). Expression of *Ms2* could simply be specific to a certain tissue or developmental stage for which there is no transcriptomic sample available for the moment. So, my feeling is that the resurrection of a pseudogene is still a hypothesis here, and I suggest describing it as a hypothesis rather than as a demonstrated result (especially as written in the title). The gene might be simply silent in the anther and the consequence of the TRIM insertion is the activation of transcription in a tissue where the gene is usually silent.

Response:

Thanks for your suggestions. We agree with your concern and we have changed the title to ‘A TRIM

insertion in the promoter of *Ms2* causes male sterility in wheat' and emphasized more about the TRIM's role in the over-expression of the downstream gene which leads to the severe phenotype of the *ms2* mutant in our revised text. Since the *Ms2* cannot be unambiguously characterized as a pseudogene, we now refer to this process as the activation of a 'dying gene' by a TRIM insertion.

Comment 8

- **TRIM mediated-expression:** I think additional details could have been provided regarding the expression of this silent gene from a TRIM. Since RNASeq data are available, could you please comment on the transcript that is generated here? Is the transcript expressed from the promoter in the LTR of the element (=chimeric transcript TE-gene)? Or is the TSS (transcription start site) outside the TRIM?

Response:

As suggested by the reviewer that we checked our RNA-Seq data, and there is no chimeric transcript or the transcripts contain TSS. To further address this possibility, we generated an artificial wheat genome sequence (Chinese Spring) with this TRIM sequence inserted in front of gene no.14 and mapped all of the RNA-seq reads from the anther libraries of the *ms2* mutant to this artificial genome sequence. No reads were mapped to this TRIM region or to any position upstream of the TSS, indicating that this TRIM insertion does not affect the gene structure of no.14 and that no chimeric gene product is produced. We have added a figure 3 in the supplementary file to display this result.

Comment 9

- **Methods: Identification of the TRIM:**

LTR_Finder is a good program to predict the presence and structure of LTR-RTs. But the predictions are often incorrect in the wheat genome especially because of nested insertions that can split LTRs into several pieces. It requires manual curation to validate the borders of the element (using dotter for instance).

Response:

We are sorry for not giving detail descriptions of the TRIM detection method. In addition to LTR_FINDER, we also used dotter (<http://nebc.nerc.ac.uk/bioinformatics/docs/dotter.html>) and manually inspected the boundaries and structure of the TRIM. The two LTRs were aligned using MEGA6 software. The predicted insertion date of a given TRIM element was estimated using the Kimura 2-parameter model with a mutation rate of 1.3×10^{-8} substitutions/site/year. We have added these details in the revised method part.

Comment 10

- **Figure 1: TDR should be corrected to LTR.**

Response:

In the previous submitted version, we used ‘TDR’ to refer to the description of TRIM’s structure according to Witte *et al.*’s paper (Proc Natl Acad Sci USA, 2001, 98:13778–83). The LTR appears in description of TRIM structure in Gao *et al.*’s paper (Genome Biology, 2016, 17:7), they prefer to use LTR instead of TDR. In the new revised version, we accept the suggestion that use LTR instead of TDR.

REVIEWERS' COMMENTS:

Reviewer #3 (Remarks to the Author):

Thank you for the point by point response to my remarks. There I found all answers to my questions. We finally agree on the data interpretation and evolutionary scenario so I have no additional comments regarding the results. I find the manuscript was significantly improved in this revised version.